# Yellow-Green and Blue Fluorescent 1,8-Naphthalimide-Based Chemosensors for Metal Cations

**Ivo Grabchev** [1,*] , **Silvia Angelova** [2] and **Desislava Staneva** [3]

1. Faculty of Medicine, Sofia University "St. Kliment Ohridski", 1407 Sofia, Bulgaria
2. Institute of Optical Materials and Technologies "Acad. J. Malinowski", Bulgarian Academy of Sciences, 1113 Sofia, Bulgaria
3. Department of Textile, Leather and Fuels, University of Chemical Technology and Metallurgy, 1756 Sofia, Bulgaria
* Correspondence: i.grabchev@chem.uni-sofia.bg; Tel.: +359-2-8161319

**Abstract:** Two new 1,8-naphthalimides (NI), emitting yellow-green or blue fluorescence depending on the type of substituents at the C-4 position, have been synthesized and characterized. Their basic photophysical characteristics have been investigated in organic solvents of different polarities. Their ability to detect metal ions ($Ag^+$, $Cu^{2+}$, $Zn^{2+}$ $Ca^{2+}$, $Mg^{2+}$, $Ni^{2+}$, and $Fe^{3+}$) has been studied in DMF solution. The amino (NI1) and alkoxy (NI2) functionalized 1,8-naphthalimides exhibit different optical and metal ion sensing properties attributable to the nature of the C-4 substituents. In addition, theoretical calculations based on the affordable but effective density functional theory (DFT) and time-dependent DFT (TDDFT) methods were performed in order to investigate the geometric and electronic structure of the title NI compounds.

**Keywords:** 1,8-naphthalimide; fluorescent sensor; PET; DFT; metal ion sensing

## 1. Introduction

Fluorescent probes have a myriad of uses in chemistry, biology, and the chemistry–biology interface. They are employed to discover new drugs, detect environmental pollutants, and most exciting, study cell biology by detecting protein location and activation, identifying protein complex formation, conformational changes, and even monitoring biological processes in vivo [1]. Popular and historically common small-molecule fluorophores for probe design are fluorescein, rhodamine, coumarin, cyanine, and BODIPY dyes. In recent years, 1,8-naphthalimides have become popular as fluorescent probes due to their excellent absorption and fluorescence properties. The core fragment is compact but can be easily modified by introducing active functional groups that can undergo further chemical transformations. The presence of a strong electron-donating (typically alkoxy or alkylamine) group at the C-4 position is a prerequisite for excellent fluorescent properties of the 1,8-naphthalimide derivative [2,3]. The introduction of an amino group to the core is a synthetically versatile method and the most used building block is the 4-amino-1,8-naphthalimide fragment [4–7]. In addition to chemical modifications, the properties of NI fluorophores are known to be highly sensitive to the polarity of the surrounding environment [4,8]. Alkoxy and amino-substituted 1,8-naphthalimides are used as dyes for textile and polymeric materials due to their very good dyeing properties [9]. The presence of polymerizing groups in the structure of 1,8-naphthalimides allows their use in the synthesis of fluorescent polymers. Thus, they are covalently linked to the main polymer chain, which prevents their migration from the polymer surface, and the resulting fluorescent effect is long-lasting [10,11].

The ability to change the photophysical characteristics by modifications of the chromophore system makes these fluorophores ideal as fluorescent units in the design of optical sensors for metal ions and protons [12–17]. Their sensitivity can be enhanced after incorporating them at the periphery of the dendrimers [18,19]. The 1,8-naphthalimide-modified

fluorescent polymers and dendrimers have been studied in the design of systems, working as heterogeneous sensors for the detection of various analytes [20].

In this work, we describe the synthesis, photophysical characteristics, and functional properties of two new functionalized 1,8-naphthalimide derivatives that possess interesting properties, including high sensitivity to solvent effects and a good ability to bind metal cations.

## 2. Results

### 2.1. Synthesis of Compounds NI1 and NI2

The initial compound NI0 has been obtained by reacting 4-nitronaphthalic anhydride with 2-aminoethylmetacrylatehydrochloride in an ethanol solution [21]. The presence of the methacrylic group in the structure of 1,8-naphthalimides leads to an expansion of their field of application. Through it, they are able to participate in copolymerization processes with traditional monomers, resulting in fluorescent polymers, in which 1,8-naphthalimides is covalently linked to the main polymer chain. In this way, the obtained polymers will have resistant color and fluorescence to wet processing and organic solvents [9]. Nucleophilic substitution of the nitro group with *N,N*-dimethylaminoethylamine (NI1) was performed in DMF at room temperature for 24 h. To obtain alkoxy-substituted naphthalimide (NI2), its bromine derivative (2) has been used. It has been prepared analogously to NI0, but 4-bromonaphthalic anhydride was used. We failed to obtain the target product (NI2) by this synthetic scheme since the ester bond was cleaved upon the addition of NaOH and the corresponding hydroxyl derivative was obtained. For this reason, we have used a nucleophilic substitution reaction of the nitro group from NI0 with *N,N*-dimethylethanolamine by ultrasonication. The reaction was carried out at 25 °C for 60 min, with the resulting product (NI2) obtained in almost quantitative yield.

*N,N*-dimethylaminoethylamine and *N,N*-dimethylethanolamine were chosen as receptor fragments for the synthesis of NI1 and NI2 since they present a tertiary nitrogen atom separated from the chromophore system by an ethylene spacer, which favors the photoinduced electron transfer [22,23].

### 2.2. Photophysical Characterization in Different Organic Solvents

The basic photophysical characteristics (absorption and emission wavelengths, respectively, $\lambda_A$ and $\lambda_F$, molar absorptivity $\varepsilon$, Stokes shifts $\nu_A - \nu_F$, and quantum yields $\Phi_F$) of 1,8-naphthalimide derivatives NI1 and NI2 have been studied in eight organic solvents of different polarities. As can be seen in Scheme 1, the difference in their chemical structure is in the part of the binding element at C-4 of the receptor fragment. The different electron-donor capacity of the oxygen and nitrogen atoms gives the compounds different photophysical and functional characteristics. This can be explained by the change in the polarization of the chromophore system as a result of the donor–acceptor interaction between the electron donor substituent at the C-4 position and the imide electron acceptor groups. From the obtained data, it can be seen that NI1 absorbs in the visible spectral region with maxima at 420–436 nm and emits yellow-green fluorescence with $\lambda_F$ = 500–538 nm. In the case of compound NI2, having an alkoxy substituent on C-4, the position of its absorption and fluorescence maxima are hypsochromically shifted. They are in the UV region at $\lambda_A$ = 358–369 nm, and the emitted fluorescence is blue with $\lambda_F$ = 410–439 nm (Tables 1 and 2).

The obtained results show that the absorption and fluorescence maxima of the two compounds depend on the polarity of the medium. Furthermore, the type of substituent significantly affects the polarization of the 1,8-naphthalimide structure and hence the position of the respective maxima. The dependence of the absorption and fluorescence maxima on the polarity of the solvents for compound NI1 is shown in Figure 1 with well-defined positive solvatochromism (a bathochromic shift with increasing solvent polarity). The solvatochromic studies are performed using the empirical Reichardt's $E_T(30)$ parameter of the solvent polarity [24]. In this case, the change in the position of the maxima depends

significantly on the polarity of the medium and the specific fluorophore-solvent interactions. A similar dependence has been observed for NI1. For both compounds, the molar extinction coefficient ($\varepsilon$) is high and it is in the range 12,400–14,400 L mol$^{-1}$ cm$^{-1}$ for NI1 and 11,900–13,900 L mol$^{-1}$ cm$^{-1}$ for NI2, which is typical of 1,8-naphthalimide derivatives. The Stokes shift ($\nu_A - \nu_F$) is in the range 3542–4342 cm$^{-1}$. The type of substituent at C-4 does not significantly affect this parameter. Since it characterizes the properties of the compounds in the ground ($S_0$) and excited ($S_1$) states, it can be concluded that both compounds retain the planarity of their core 1,8-naphthalimide structure.

**Scheme 1.** Synthesis of compounds NI1 and NI2.

**Table 1.** Photophysical characteristics of compound NI1 in different organic solvents.

| Solvent | $\lambda_A$ nm | $\varepsilon$ L mol$^{-1}$ cm$^{-1}$ | $\lambda_F$ nm | $\nu_A - \nu_F$ cm$^{-1}$ | $\Phi_F$ |
|---|---|---|---|---|---|
| Acetonitrile | 436 | 12,400 | 517 | 3593 | 0.022 |
| *N,N*-dimethylformamide | 429 | 12,900 | 525 | 4262 | 0.010 |
| n-Butanol | 434 | 12,600 | 525 | 3994 | 0.018 |
| Ethanol | 438 | 12,700 | 536 | 4174 | 0.011 |
| Methanol | 439 | 12,400 | 538 | 4192 | 0.010 |
| Chloroform | 423 | 14,300 | 502 | 3720 | 0.716 |
| Dichloromethane | 424 | 14,600 | 504 | 3753 | 0.462 |
| Tetrahydrofuran | 420 | 14,400 | 500 | 3809 | 0.894 |

**Table 2.** Photophysical characteristics of compound NI2 in different organic solvents.

| Solvent | $\lambda_A$ nm | $\varepsilon$ L mol$^{-1}$ cm$^{-1}$ | $\lambda_F$ nm | $\nu_A - \nu_F$ cm$^{-1}$ | $\Phi_F$ |
|---|---|---|---|---|---|
| Acetonitrile | 366 | 12,300 | 422 | 3625 | 0.021 |
| *N,N*-dimethylformamide | 367 | 11,900 | 426 | 3773 | 0.015 |
| n-Butanol | 365 | 12,000 | 425 | 3867 | 0.019 |
| Ethanol | 368 | 12,200 | 438 | 4342 | 0.008 |
| Methanol | 369 | 12,400 | 439 | 4321 | 0.007 |
| Chloroform | 360 | 13,900 | 414 | 3623 | 0.624 |
| Dichloromethane | 360 | 13,600 | 413 | 3564 | 0.762 |
| Tetrahydrofuran | 358 | 13,400 | 410 | 3542 | 0.777 |

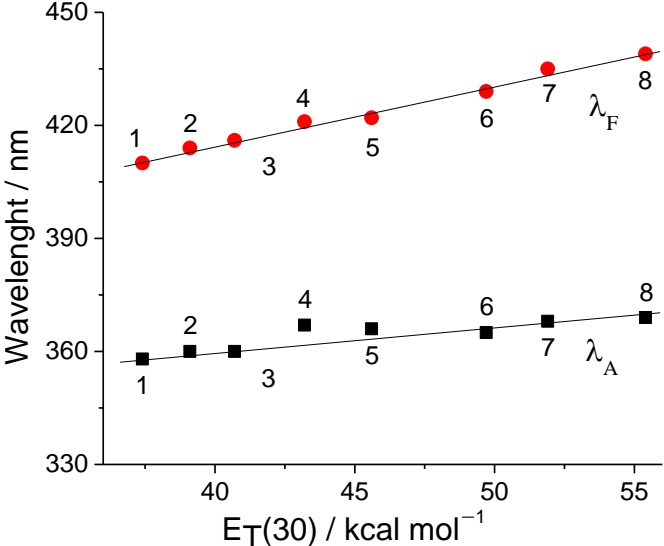

**Figure 1.** Dependence of absorption (black squares) and fluorescence (red circles) maxima of NI2 on the polarity of organic solvents ($E_T(30)$): (1) tetrahydrofuran, (2) chloroform, (3) dichloromethane, (4) *N,N*-dimethylformamide, (5) acetonitrile, (6) *n*-butanol, (7) ethanol, and (8) methanol.

Figure 2 shows the dependence of the quantum yield of NI1 and NI2 on the polarity of the solvents. Both compounds show much lower intensity in polar media compared with apolar ones. The fluorescence is amplified in apolar media more than 110 times for NI2 and 89 times for NI1. Due to the presence of the electron-donating alkoxy and alkylamino groups at the C-4 atom and their conjugation with the electron-accepting carbonyl groups, the long wavelength absorption bands have a charge transfer character (CT). These bands are sensitive to the polarity of the solvents, whereby internal charge transfer (ICT) is observed. These electronic interactions are, on the one hand, responsible for the shift of the absorption and emission spectral bands and, on the other hand, affect



the fluorescence quantum yield. Dipole–dipole interactions in polar solvents additionally decrease the polarization of the chromophore system, reducing the quantum yield of NI1 and NI2 in polar solvents. The results obtained are very different than the ones obtained for alkoxy and amino-substituted 1,8-naphthalimides without (-$CH_2CH_2N(CH_3)_2$) receptor fragments [25,26].

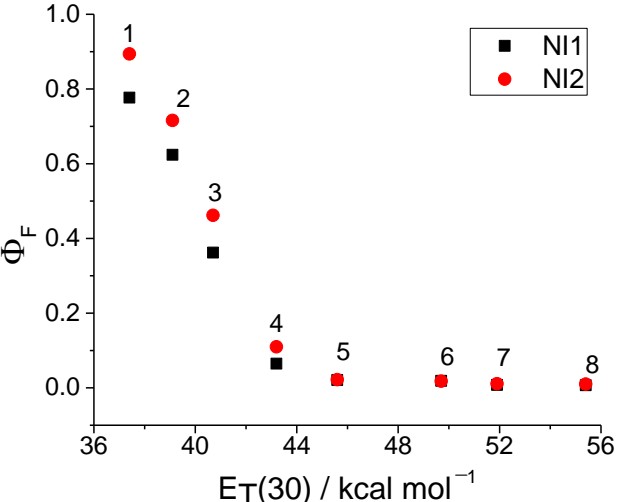

**Figure 2.** Dependence of quantum yield of NI1 and NI2 on the polarity of organic solvents: (1) tetrahydrofuran, (2) chloroform, (3) dichloromethane, (4) *N*,*N*-dimethylformamide, (5) acetonitrile, (6) *n*-butanol, (7) ethanol, and (8) methanol.

In the molecules of NI1 and NI2, a presumed receptor fragment is the dimethylamino group, which is attached to the 1,8-naphthalide structure via ethylamino (NI1) or ethyloxy (NI2) spacer. This receptor fragment contains a tertiary nitrogen atom with an unshared electron pair capable of protonation or forming complexes with metal ions. When light is absorbed, photoinduced electron transfer (PET) takes place, which quenches the fluorescence of the 1,8-naphthalimide signal fragment and the emission of the sensor is "*off*". In the presence of analytes (protons or metal cations), as a result of their interaction with the free electron pair from the receptor fragment, PET is extinguished and the fluorescence emission is restored.

In order to clarify the sensor activity of NI1 and NI2, their photophysical characteristics have been studied in the presence of metal ions with different charges: $Ag^+$, $Cu^{2+}$, $Zn^{2+}$, $Ca^{2+}$, $Mg^{2+}$, $Ni^{2+}$, and $Fe^{3+}$ (nitrate salts of the metal ions were used). Since compounds NI1 and NI2 are insoluble in water, we investigated their sensory ability in DMF (a good solvent for both the ligands and the resulting complexes). In addition, in DMF solution, the emitted fluorescence by both compounds is low due to PET processes.

When studying the influence of metal ions on photophysical characteristics, it was found that the fluorescence intensity increased, and the increase depends on both the nature of the metal ion and the substituent in C-4. To quantify the increase in fluorescence, the fluorescence enhancement factor (FE = $I/I_0$) was used, which has been calculated by the ratio of the fluorescence intensity in the presence of metal ions (I) and the initial intensity in the absence of metal ions ($I_0$). The results obtained for FE of NI1 and NI2 in the presence of $Ag^+$, $Cu^{2+}$, $Zn^{2+}$, $Ca^{2+}$, $Mg^{2+}$, $Ni^{2+,}$ and $Fe^{3+}$ are shown in Figure 3.

It can be seen that the highest value for FE was obtained in the presence of $Fe^{3+}$ for both compounds, followed by copper, nickel, and silver ions. The slightest increase in fluorescence is caused by calcium and magnesium ions. Probably the most stable metal complex bound to the *N*,*N*-dimethylamino receptor fragment is obtained with $Fe^{3+}$. These results are also confirmed by the quantum yields of fluorescence of the compounds in a solution of DMF in the presence of the ions (Tables 3 and 4). Additionally, from the data, it can be seen that the position of the absorption maxima of the compounds is

slightly influenced by the metal ions. A significantly stronger effect has been observed in the fluorescence spectra, the maxima of which are hypsochromically shifted in the presence of those metal ions which cause an increase in fluorescent intensity, and this effect is particularly pronounced for $Fe^{3+}$ ($\Delta\lambda_F$ = 13–15 nm). That means that the metal complexes with 1,8-naphthalimide ligands are formed mainly in the receptor fragment of the substituents at the C-4 atom. The obtained results show that despite the good ability to recognize metal ions in non-aqueous media, they do not have specific selectivity for a particular ion.

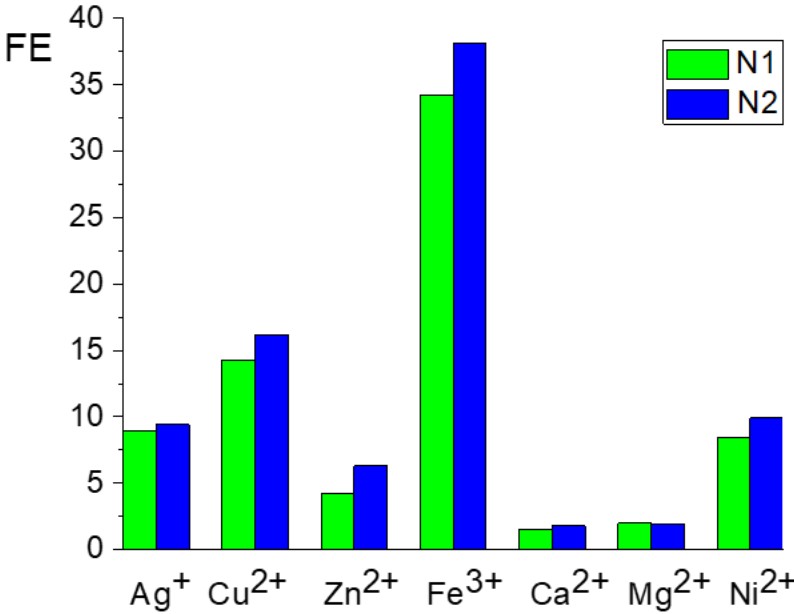

**Figure 3.** Fluorescence enhancement factor (FE) of NI1 and NI2 (c = 1 × $10^{-5}$ mol $L^{-1}$) in the presence of metal cations (c = 1 × $10^{-5}$ mol $L^{-1}$) in DMF solution.

**Table 3.** Photophysical characteristics of NI1 in DMF solution and presence of metal cations.

| Metal Ions | $\lambda_A$/nm | $\lambda_F$/nm | FE | Stokes Shift/$cm^{-1}$ | $\Phi_F$ |
|---|---|---|---|---|---|
| $Ag^+$ | 428 | 515 | 8.90 | 3947 | 0.103 |
| $Cu^{2+}$ | 429 | 516 | 14.29 | 3930 | 0.169 |
| $Zn^{2+}$ | 429 | 520 | 4.20 | 4079 | 0.048 |
| $Fe^{3+}$ | 428 | 510 | 34.25 | 3756 | 0.384 |
| $Ni^{2+}$ | 429 | 521 | 8.40 | 4116 | 0.101 |
| $Ca^{2+}$ | 428 | 524 | 1.5 | 4280 | 0.016 |
| $Mg^{2+}$ | 428 | 524 | 1.9 | 4280 | 0.017 |

**Table 4.** Photophysical characteristics of NI2 in DMF solution and presence of metal cations.

| Metal Ions | $\lambda_A$/nm | $\lambda_F$/nm | FE | Stokes Shift/$cm^{-1}$ | $\Phi_F$ |
|---|---|---|---|---|---|
| $Ag^+$ | 365 | 416 | 9.40 | 3358 | 0.151 |
| $Cu^{2+}$ | 365 | 415 | 16.23 | 3301 | 0.249 |
| $Zn^{2+}$ | 367 | 422 | 6.32 | 3551 | 0.101 |
| $Fe^{3+}$ | 365 | 412 | 38.05 | 3125 | 0.571 |
| $Ni^{2+}$ | 366 | 415 | 9.9 | 3226 | 0.147 |
| $Ca^{2+}$ | 365 | 425 | 1.8 | 3867 | 0.027 |
| $Mg^{2+}$ | 365 | 426 | 1.9 | 3933 | 0.028 |

As a typical example, Figure 4 shows the fluorescence spectra of NI2 ($c = 10^{-5}$ mol $L^{-1}$) upon the addition of different concentrations of $Fe^{3+}$ ions. It can be seen that with an increase in their concentration up to $10^{-5}$ mol $L^{-1}$, the fluorescence intensity increases, after which it remains constant. Figure 4 also shows the hypsochromic shift of the position of the fluorescent maximum in the presence of $Fe^{3+}$ ions. The inset shows that the stoichiometry of the complex formed between $Fe^{3+}$ ions and 1,8-naphthalimide is 1:1. Job's plot has also been used to determine the binding stoichiometry of the NI2 with $Fe^{3+}$ ions (Figure 5). It confirmed that the binding stoichiometry is 1:1.

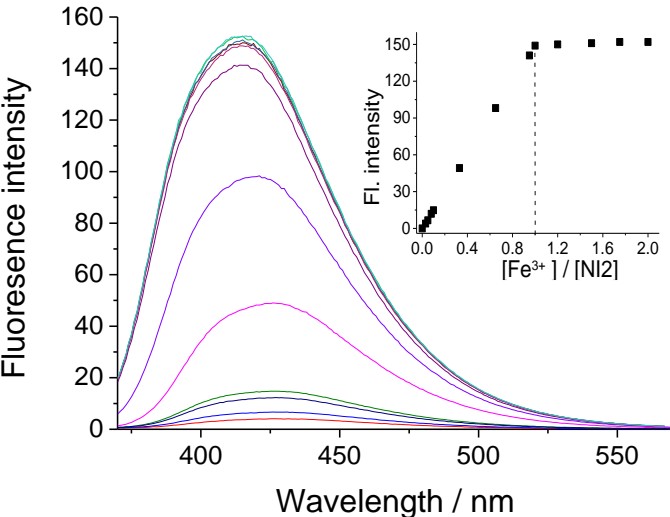

**Figure 4.** Fluorescence spectra of NI2 ($c = 1 \times 10^{-5}$ mol $L^{-1}$) at various concentrations of $Fe^{3+}$ cations (from 0 to $1.5 \times 10^{-5}$ mol $L^{-1}$).

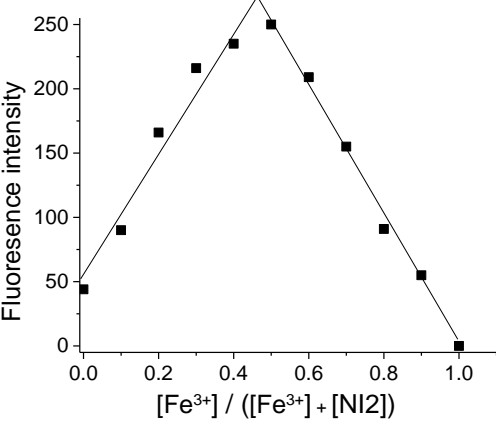

**Figure 5.** Job's plot for the $Fe^{3+}$ complex of NI2.

At the concentration of $Fe^{3+}$ ions in the range $c = 0$–$9 \times 10^{-6}$ mol $L^{-1}$, a very good linear dependence was obtained, with R = 0.9999 (Figure 6). Based on the linear regression, the limit of detection LOD = $4.48 \times 10^{-7}$ mol $L^{-1}$ and the limit of quantification LOQ = $1.49 \times 10^{-6}$ mol $L^{-1}$ were calculated by the following dependencies: LOD = 3Sa/b and LOQ = 10Sa/b, where Sa is the standard deviation of the response and b is the slope of the calibration curve [27]. The values obtained are smaller than the ppm concentration range and are low enough for the detection of $Fe^{3+}$. A similar dependence was obtained when using NI1 as a ligand.

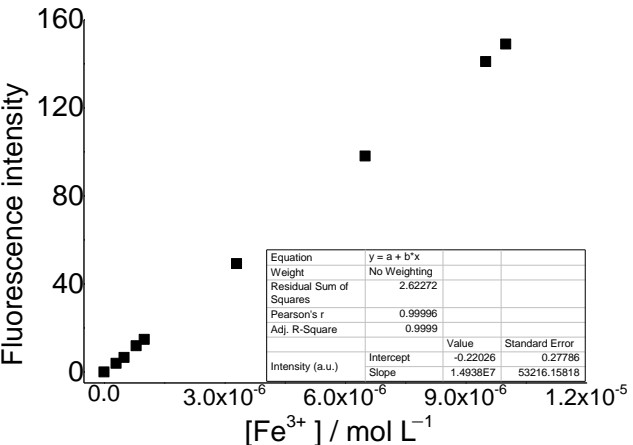

**Figure 6.** Linear calibration curve of NI2 in DMF solution (c = $1 \times 10^{-5}$ mol L$^{-1}$) at various concentrations of Fe$^{3+}$ cations (from 0 to $1 \times 10^{-5}$ mol L$^{-1}$) in the order of increasing intensity (λex = 365 nm).

### 2.3. Computational Studies

To gain insight into the geometric and electronic structure of the new NI1 and NI2 naphthalimides, we have investigated theoretically the title compounds and their metal complexes by means of DFT and TDDFT calculations. As a first step, a full geometry optimization of the molecular structures was performed at the B3LYP/6-31+G(d,p) level of theory in the gas phase (ε = 1.0), toluene (ε = 2.4), chloroform (ε = 4.7), methanol (ε = 32.6), *N*,*N*-dimethylformamide (37.2), and water (ε = 78.4) solvent environments. Gas phase optimized structures of the low-energy isomers of NI1 and NI2 are visualized in Figure 7. The dyes are structurally similar, with the same substituent on the imide nitrogen atom, and differing only at the C-4 substituent, *N*,*N*-dimethylethyl-substituted amino, or alkoxy group. Hirshfeld charges on selected heteroatoms obtained for the optimized NI1 and NI2 geometries are shown in Figure 7.

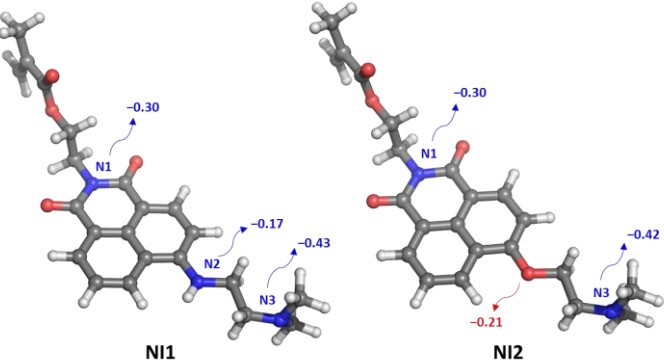

**Figure 7.** B3LYP-optimized structures of NI1 and NI2. Hirshfeld charges on selected atoms were calculated for NI1 and NI2.

Despite the fact that the dyes are structurally similar, their electronic characteristics are different, and the difference originates from the C-4 amino/alkoxy-1,8-naphthalimide segment. The character of the transitions and the influence of the C-4 substituents on the spectral properties were studied by density functional theory (TDDFT) calculations at the TDPBE0/6–311+G(2d,p)//B3LYP/6-31+G(d,p) level of theory. The experimentally measured absorption wavelengths (Table 1) depend on the nature and the polarity of the organic solvents used, thus the solvent effects need to be accounted for in the prediction of the spectral properties of NIs. The computational approach (PCM) used in the study of the NI solvatochromic probes is a popular choice of implicit solvent model. The combination of PCM and TDDFT allows the experimentally observed spectroscopic trends to be

qualitatively reproduced. The predicted absorption wavelengths of the lowest electronic transitions and oscillator strengths for NI1 and NI2 are listed in Table 5 and visualized in Figure S1. The dipole moments are found to gradually increased when going from non-polar to polar solvents. NI1 has a greater dipole moment compared to NI2 in the respective solvent.

**Table 5.** TDDFT calculated absorption wavelengths ($\lambda_{abs}$, nm), oscillator strength f (in parentheses), and dipole moments ($\mu$, Debye) for NI1 and NI2 in solvents of different polarity: toluene, chloroform, methanol, DMF, and water.

| Solvent | NI1 | | | NI2 | | |
|---|---|---|---|---|---|---|
| | calc. | exp. | $\mu$ | calc. | exp. | $\mu$ |
| Toluene | 411 (0.34) | - | 9.16 | 364 (0.35) | - | 7.28 |
| Chloroform | 415 (0.34) | 423 | 9.99 | 366 (0.35) | 360 | 7.71 |
| Methanol | 419 (0.33) | 439 | 10.87 | 368 (0.34) | 369 | 8.18 |
| DMF | 420 (0.34) | 429 | 10.89 | 369 (0.35) | 367 | 8.19 |
| Water | 419 (0.33) | - | 10.97 | 368 (0.34) | - | 8.23 |

The first excited states are determined by HOMO → LUMO (for NI1) and HOMO-1 → LUMO (for NI2) transitions with oscillator strengths f ≈ 0.35. The frontier orbitals of the studied compounds are shown in Figure 8. HOMO orbitals are located differently for NI1 and NI2: entirely on the 1,8-naphtalimide core for NI1 and on the dimethylamino group for NI2. LUMOs are delocalized on the 1,8-naphtalimide core for both compounds. The differences in the spatial distribution of the frontier orbitals explain the experimentally observed differences in the photophysical properties of NI1 and NI2.

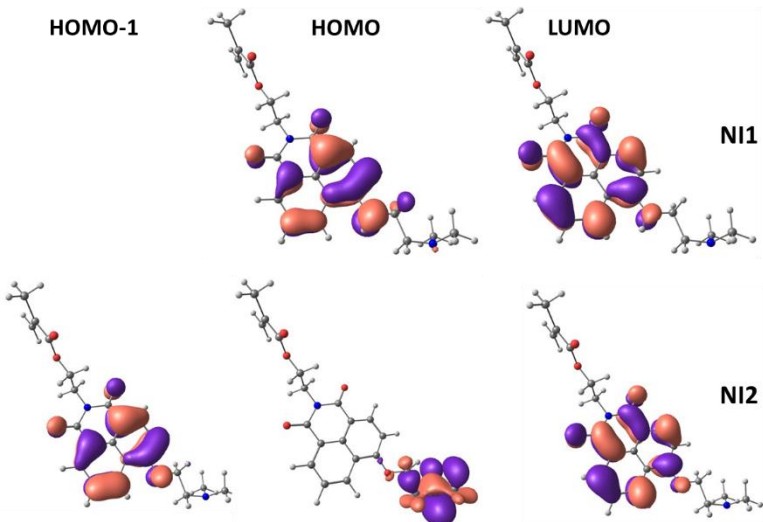

**Figure 8.** Graphical representation of the frontier orbitals (isodensity plot, isovalue = 0.02 a.u.).

It should be noted that the implicit solvent model used treats solvent as a continuous medium and does not differentiate protic (methanol and water) from aprotic (DMF) polar solvents ($\varepsilon > 15$).

To probe the dominant protonation sites on NI1 (N1, N2, and N3 atoms) and NI2 (N1, O, and N3 atoms) structures (Figure 7), we compared the stabilities of the different protonated forms (Table 6). As expected, the N3-H$^+$ form (with protonated dimethylamino group) appears to be the most stable one for both NI1 and NI2 compounds.

**Table 6.** Gibbs energy differences ($\Delta G$) between the protonated forms of NI1 and NI2 in the gas phase ($\Delta G^1$) and water ($\Delta G^{78}$), in kcal mol$^{-1}$.

| Protonation Site | NI1 | | NI2 | |
|:---:|:---:|:---:|:---:|:---:|
| | $\Delta G^1$ | $\Delta G^{78}$ | $\Delta G^1$ | $\Delta G^{78}$ |
| N1 | 18.7 | 42.5 | 22.1 | 42.9 |
| N2/O | 10.9 | 20.9 | 34.8 | 43.9 |
| N3 | 0.0 | 0.0 | 0.0 | 0.0 |

Two types of NI-metal complexes were modeled by placing metal ions ($Fe^{3+}$ and $Mg^{2+}$) close to (1) the imide nitrogen atom substituent and (2) the receptor dimethylamino group, allowing the system to relax. As mentioned above, FE differs significantly in the presence of $Fe^{3+}$ and $Mg^{2+}$ ions. The highest increase in fluorescence was observed in the presence of $Fe^{3+}$ ions for both compounds, while the slightest increase was in the presence of $Mg^{2+}$ and $Ca^{2+}$ ions. The B3LYP optimized structures of the modeled iron complexes are shown in Figure 9; $Mg2^+$ ions are similarly positioned in the corresponding NI1/NI2-magnesium complexes.

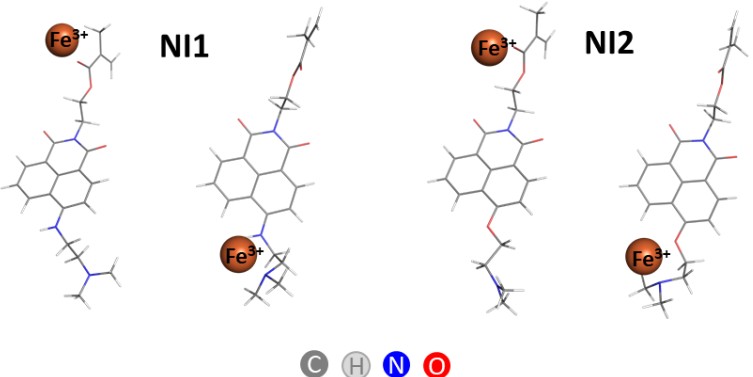

**Figure 9.** B3LYP optimized structures of NI1 and NI2 complexes with $Fe^{3+}$ ions.

The negative values of the Gibbs energies calculated for the complex formation reactions between $Fe^{3+}$ ions and NIs indicate spontaneous and energy-favorable processes in the DMF environment with a preference for the N-substituent binding site (Table 7): $-107.7$ kcal mol$^{-1}$ vs. $-98.4$ kcal mol$^{-1}$ for NI1 and $-102.7$ kcal mol$^{-1}$ vs. $-93.4$ kcal mol$^{-1}$, respectively. The binding between the NI ligands and $Mg^{2+}$ is weaker. $Mg^{2+}$ tends to coordinate preferably with the C-4 amino/alkoxy fragment ($-7.6$ kcal mol$^{-1}$ and $-8.7$ kcal mol$^{-1}$ for NI1 and NI2, respectively); $Mg^{2+}$ coordinates only with the imide nitrogen fragment of NI1 ($-3.5$ kcal mol$^{-1}$ and 7.0 kcal mol$^{-1}$ for NI1 and NI2, respectively).

**Table 7.** Gibbs energies $\Delta G^{37}$ for the formation of metal complexes (reactions NI + $M^{2+/3+} \rightarrow$ NI@$M^{2+/3+}$) in DMF, in kcal mol$^{-1}$.

| Position | $Mg^{2+}$ | $Fe^{3+}$ |
|:---:|:---:|:---:|
| NI1 (N substituent) | $-3.5$ | $-107.7$ |
| NI1 (C-4 substituent) | $-7.6$ | $-98.4$ |
| NI2 (N substituent) | 7.0 | $-102.7$ |
| NI2 (C-4 substituent) | $-8.7$ | $-93.4$ |

According to the DFT modeling results, the magnesium complexes with 1,8-naphthalimide ligands are formed mainly (for NI1 and only for NI2) in the receptor fragment of the substituents at the C-4 atom. The binding affinity of NIs for $Fe^{3+}$ is much stronger, as confirmed by the predicted high negative Gibbs energies for the complex formation reaction between $Fe^{3+}$ ions and NIs. Coordination of $Fe^{3+}$ with two binging sites is possible.

## 3. Materials and Methods

### 3.1. Synthesis of 1,8-Naphthalimides

The initial 2-(6-nitro)-1,3-dioxo-1H-benzo[de]isoquinolin-2(3H)-yl)ethyl methacrylate (NI0) has been synthesized by the method described recently [21].

### 3.2. Synthesis of 2-(6-(2-(N,N-Dimethylaminoethylamino)-1,3-dioxo-1H-benzo[de]isoquinolin-2(3H)-yl)ethyl Methacrylate (NI1)

*N,N*-dimethylaminoethylamine (0.096 mL, 0.15 55 mmol) was added to NI0 (0. 354 g, 0.1 mmol) dissolved in 20 mL DMF and the mixture was stirred 24 h at 30 °C. The reaction was monitored by TLC (1:1 n-hexane:acetone). At the end of the reaction, the liquor was added to 200 mL of ice water. The precipitated was filtrated, washed with water, and dried. The yield was 0.367 g, 89%.

FTIR cm$^{-1}$: 2952, 1717, 1684, 1638, 1579, 1457, 1364, 1244, 1165, 1057, 775, and 758.

$^1$H-NMR (DMSO-d$_6$, 600 MHz, δ (ppm): 8.64 (d, J = 7.3 Hz, 1H, Ar-H), 8.42 (d, J = 7.2 Hz, 1H, Ar-H), 8.26 (d, J = 8.0 Hz, 1H, Ar-H), 7.68 (m, 2H, Ar-H + N-H), 6.80 (d, J = 8.0 Hz, 1H, Ar-H), 5.91 (s, 1H trans=CHH), 5.57 (s, 1H cis=CHH), 4.36 (m, 2H, NCH$_2$CH$_2$O), 4.11 (m, 2H, NCH$_2$CH$_2$O), 3.09 (br.s, 2H, NHCH$_2$CH$_2$N<), 2.71 (br.s, 2H, NHCH$_2$CH$_2$N<), 2,32 (s, 6H, N-CH$_3$), and 1.76 (s, 3H, *CH$_3$*-C=CH$_2$).

$^{13}$C NMR (DMSO-d$_6$, 150 MHz, δ (ppm): 173.3, 164.9, 163.8, 149.9, 136.4, 134.2, 133.0, 132.2, 126.9, 125.3, 124.0, 120.6, 115.8, 107.1, 103.8, 75.8, 61.9, 56.6, 45.8, 43.4, and 17.9.

Elemental analysis: C$_{22}$H$_{25}$N$_3$O$_4$ (395.2) Calc: C-66.80; H-6.33; N-10.62 Found: C-66.61; H-6.28; N-10.56.

### 3.3. Synthesis of 2-(6-(2-(N,N-Dimethylaminoethoxy)-1,3-dioxo-1H-benzo[de]isoquinolin-2(3H)-yl)ethyl Methacrylate (NI2)

A total of 0.354 g (0.1 mmol) NI0 was dissolved in 20 mL *N,N*-dimethylethanolamine and (0.1 g, 1.0 mmol) K$_2$CO$_3$ was added. The reaction was carried out in an ultrasonic bath (125 W/25 Hz) at 25 °C for 60 min. The chemical reaction was monitored by TLC (1:1 n-hexane:acetone). After that, the solvent was removed by vacuum and the solid residue was dissolved in 50 mL chloroform and washed with water. The organic phase solution was separated and the chloroform was evaporated. The final product was obtained with high yield and purity. The yield was 0.403 g, 98%.

FTIR cm$^{-1}$: 2963, 1715, 1695, 1648, 1463, 1377, 1244, 1161, 1066, 775, and 758.

$^1$H-NMR (DMSO-d$_6$, 400 MHz, δ (ppm): 8.68 (d, J = 7.1 Hz, 1H, Ar-H), 8.48 (d, J = 7.8 Hz, 1H, Ar-H), 8.31 (d, J = 8.2 Hz, 1H, Ar-H), 7.72 (d, J = 7.8 Hz, 2H, Ar-H), 6.83 (d, J = 7.8 Hz, 1H, Ar-H), 5.94 (s, 1H trans=CHH), 5.63 (s, 1H cis=CHH), 4.30 (m, 2H, NCH$_2$CH$_2$O), 4.18 (m, 2H, NCH$_2$CH$_2$O), 3.19 (br.s, 2H, OCH$_2$CH$_2$N<), 2.73 (br.s, 2H, NHCH$_2$CH$_2$N<), 2,36 (s, 6H, N-CH3), and 1.77 (s, 3H, *CH$_3$*-C=CH$_2$).

$^{13}$C NMR (DMSO-d$_6$, 150 MHz, δ (ppm): 167.6, 164.8, 164.5, 150.8, 137.9, 136.4, 134.0, 130.1, 125.8, 124.9, 120.2, 115.3, 109.0, 108.0, 104.3, 67.3, 62.8, 60.1, 59.2, 45.9, and 16.1. Elemental Analysis: C$_{22}$H$_{24}$N$_2$O$_5$ (396.2) Calc: C-66.63; H-6.06; N-7.07 Found: C-66.52; H-6.08; N-7.16.

### 3.4. Synthesis of 2-(6-bromo)-1,3-dioxo-1H-benzo[de]isoquinolin-2(3H)-yl)ethyl Methacrylate

Trimethylamine (1.53 mL) was added to a 20 mL ethanol solution of 2-aminoethylmeta crylate hydrochloride (1.82 g, 0.011 mol) and the mixture was vigorously stirred for 30 min. This solution was added to the ethanol solution of 4-bromonaphthalic anhydride (2.81 g, 0.01 mol) at 60 °C for 2 h. After cooling to room temperature, the precipitate was filtered, washed with water, and dried in a vacuum at 40 °C. The yield was 2.98 g, 84%.

FTIR cm$^{-1}$: 2948, 1721, 1698, 1653, 1469, 1358, 1244, 1163, 1069, 777, and 760.

1H-NMR (DMSO-d$_6$, 400 MHz, δ (ppm): 8.70 (d, J = 7.9 Hz, 1H, Ar-H), 8.52 (d, J = 7.8 Hz, 1H, Ar-H), 8.46 (d, J = 8.3 Hz, 1H, Ar-H), 8.00 (m, 1H, Ar-H), 7.7 (t, J = 6.90 Hz, 1H, Ar-H), 6.10 (s, 1H trans=CHH), 5.50 (s, 1H cis=CHH), and 4.50 (dd, J = 5.96 Hz 4H, NCH$_2$CH$_2$O), 1.80 (s, 3H, *CH$_3$*-C=CH$_2$).

Elemental Analysis: $C_{18}H14NO_4Br$ (388.3) Calc: C-55.62; H-3.60; N-3.60 Found: C-55.86; H-3.54; N-3.66.

### 3.5. Analysis

The UV–Vis spectrophotometric investigations were performed on a UV–Vis "Thermo Spectronic Unicam UV 500" double-beam spectrophotometer. Fluorescence spectra were taken on a "Cary Eclipse" fluorescence spectrophotometer. The absorption and fluorescence spectra were recorded using $10^{-5}$ mol/L solutions for NI1 and NI2 monomers. The quantum yield of fluorescence $\Phi_F$ has been calculated on the basis of the absorption and fluorescence spectra of NI1 or NI2 by Equation (1):

$$\Phi_F = \Phi_{st} \frac{S_u}{S_{st}} \frac{A_{st}}{A_u} \frac{n_u^2}{n_{st}^2} \tag{1}$$

where $\Phi_F$ is the emission quantum yield of the sample; $\Phi_{st} = 0.86$ is the emission quantum yield of the standard (Rhodamine 6G ($\Phi_{st} = 0.96$) and quinine bisulfate/$H_2SO_4$ 1N ($\Phi_{st} = 0.546$) for NI2); $A_{st}$ and $A_u$ represent the absorbance of the standard and sample at the excited wavelength, respectively; $S_{st}$ and $S_u$ are the integrated emission band areas of the standard and sample, respectively; $n_{st}$ and $n_u$ are the solvent refractive indexes of the standard and sample; and subscripts $u$ and $st$ refer to the unknown and standard, respectively.

The influence of the metal cations on the fluorescence intensity has been investigated by adding a few μL of stock solution of the metal cations to a known volume of the NI1 or NI2 in DMF solution (3 mL). ATR-FTIR spectroscopic analyses were performed using an IRAffinity-1 spectrophotometer (Shimadzu Co., Kyoto, Japan). $^1$H and $^{13}$C NMR spectra were acquired on a Bruker 400 MHz NMR spectrometer (Bruker BioSpin GmbH, Rheinstetten, Germany,). The measurements were carried out in a DMSO-$d_6$ solution. TLC monitoring was conducted using silica gel (Fluka $F_{60}$ 254 20 × 20; 0.2 mm) and $n$-hexane/acetone/(1:1) as an eluent.

### 3.6. DFT Computations

All calculations were performed using Gaussian 09 (G09) computational chemistry software package [28]. The molecular geometries of NI1 and NI2 naphthalimides were fully optimized at the B3LYP/6-31+G(d,p) level using the B3LYP [29,30] function and a diffuse function-augmented version of 6-31G(d,p) [31,32] Pople's basis set for all atoms. Detailed descriptions of the adopted computational protocol (including polarizable continuum model calculations [33]) and PyMOL [34] visualization of the structures are given as Supplementary Materials.

## 4. Conclusions

Two new 1,8-naphthalimide derivatives have been synthesized and characterized. Receptor fragments have been introduced into their structure, which allows the detection of metal ions. Their main photophysical characteristics were investigated in organic solvents of different polarities. It was found that they emit blue and green fluorescence depending on the nature of the C-4 substituent in the naphthalimide structure. Their quantum yield strongly depends on the polarity of the medium. The experimental sensor activity of NI1 and NI2 has been studied in DMF in presence of different metal ions—$Ag^+$, $Cu^{2+}$, $Zn^{2+}$ $Ca^{2+}$, $Mg^{2+}$, $Ni^{2+}$, and $Fe^{3+}$. The highest value for FE was obtained in the presence of $Fe^{3+}$, while the slightest increase in fluorescence was dictated by $Ca^{2+}$ and $Mg^{2+}$ ions. Theoretically predicted negative Gibbs energy values of the complex formation between $Fe^{3+}$ ions and NIs indicated spontaneous and energy-favorable complex formation processes in the DMF environment, whereas $Mg^{2+}$ affinity for NI binding sites was much lower.

**Supplementary Materials:** The following supporting information can be downloaded at: https://www.mdpi.com/article/10.3390/inorganics11020047/s1, Figure S1: 1H-NMR spsctra of NI1; Figure S2: 1C-NMR spsctra of NI1; Figure S3: 1H-NMR spsctra of NI2; Figure S4: 1C-NMR spsctra

of NI2; Figure S5: TDPBE0/6-311+G(2d,p) simulated spectra (Gaussian broadening, band width on ½ height: 0.1 eV).

**Author Contributions:** Conceptualization, I.G.; methodology, I.G., S.A. and D.S.; formal analysis, I.G., S.A. and D.S.; investigation, I.G., S.A. and D.S.; writing—original draft preparation, I.G., S.A. and D.S.; writing—review and editing, I.G., S.A. and D.S.; visualization, I.G., S.A. and D.S.; supervision, I.G.; project administration, I.G.; funding acquisition, I.G. All authors have read and agreed to the published version of the manuscript.

**Funding:** The authors gratefully acknowledge the financial support from the Bulgarian National Science Fund under grant КП-06-Н49/2.

**Data Availability Statement:** The data presented in this study are available on request from the corresponding author.

**Acknowledgments:** S.A. acknowledges the provided access to the e-infrastructure of the NCHDC—part of the Bulgarian National Roadmap on RIs.

**Conflicts of Interest:** The authors declare no conflict of interest.

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
