# Peer review of "Yellow-Green and Blue Fluorescent 1,8-Naphthalimide-Based Chemosensors for Metal Cations"

_inorganics, doi:10.3390/inorganics11020047_

Round 1

Reviewer 1 Report

The work of Grabchev and co-authors entitled "Yellow-green and blue fluorescent 1,8-Naphthalimide-based chemosensors for metal cations" reports an interesting approach to metal responsive fluorescence sensors. The paper describes the synthesis and characterization of two new 1,8-naphthalimide derivatives. These sensing molecules are capable of emitting blue and yellow-green fluorescence depending on the electron-donating ability of a substituent at the C-4 atom.  Also, their spectral properties in different solvents were investigated. The geometric and electronic structure of the synthesized compounds was investigated by density functional theory (DFT) and time-dependent DFT (TDDFT).

The ability of the new compounds to detect metal-dissimilar metals was investigated.

The article is well structured and fully corresponds to the scope of the journal. The work is well described with sound conclusions and, in my opinion, deserves to be published after some minor revisions. Also, I think the paper will be of interest to the journal readers.

Before publishing, I would like the following points to be clarified.

1. Why an unsaturated methacrylic group was introduced into the structure of the compounds

2.  Did the authors use different characteristics than the one presented ET30 in Figures 1 and 2 to clarify the dependence on the polarity of organic solvents?

Author Response

  1. Why an unsaturated methacrylic group was introduced into the structure of the compounds?

Authors’ reply: A short explanation has been added on p.4 (3.1. Synthesis of compounds NI1 and NI2): “The presence of the methacrylic group in the structure of 1,8-naphthalimides leads to an expansion of their field of application. Through it, they will be able to participate in copolymerization processes with traditional monomers in order to obtain fluorescent polymers in which 1,8-naphthalimides will be covalently bonded to the main polymer chain. In this way, the obtained polymers will have resistant color and fluorescence to wet processing and organic solvents [9].”

  1. Did the authors use different characteristics than the one presented ET30 in Figures 1 and 2 to clarify the dependence on the polarity of organic solvents?

Authors’ reply: Different solvatochromic parameters measuring the polarity, hydrogen bond donation (Lewis acidity) and electron pair acceptance (Lewis basicity) abilities can be determined spectroscopically. The chosen by us empirical parameter for solvent polarity ET(30) provides information on the dipolarity/polarizability and hydrogen bonding interactions. There is a good correlation between the absorbance/fluorescence of NIs and this empirical parameter, that is, the experimental results confirm that the (photo)chemical properties of NIs depend on the polarity of the chosen medium and its hydrogen bonding ability. The performed TDDFT calculations by an implicit solvent model (PCM) that treats solvent as a continuous medium, confirm the experimentally observed dependence on the solvent polarity.

Reviewer 2 Report

The paper by Grabchevet al. presents a new example of sensors for metal cations by photoluminescence detection. The research is incremental but adds additional information. It is suitable for publication after minor revisions:

1.    Scheme 1 needs modifying. It is long ago typically write alkane chains as sticks but not like in the scheme (CH2CH2 etc.). in addition, please correctly demonstrate both carbonyl groups in Naphthalimide fragments. Angle CCO should be closer to 120 deg, even in the scheme.

2.    All experimental data such as NMR and electronic (UV and PL) spectra should be given in supporting information.

3.    It is not required to connect points on the Job’s plot for better presentation.

4.    As a suggestion. There are several spectroscopic methods to establish the centers of coordination of metal. For example, IR or NMR spectroscopy could help to demonstrate the presence or absence of coordination of metals with the CO or NH groups. Sure, it should be performed at higher concentrations and could be used by authors further.

Author Response

  1. Scheme 1 needs modifying. It is long ago typically write alkane chains as sticks but not like in the scheme (CH2CH2 etc.). in addition, please correctly demonstrate both carbonyl groups in Naphthalimide fragments. Angle CCO should be closer to 120 deg, even in the scheme.

Authors’ reply: Scheme 1 has been modified following the Reviewer’s recommendation.

  1. All experimental data such as NMR and electronic (UV and PL) spectra should be given in supporting information.

Authors’ reply: The authors confirm that the raw data supporting the findings of this study will be available from the corresponding author, [IG], upon reasonable request.

  1. It is not required to connect points on the Job’s plot for better presentation.

Authors’ reply: Figure 5 has been modified as suggested by the Reviewer.

  1. As a suggestion. There are several spectroscopic methods to establish the centers of coordination of metal. For example, IR or NMR spectroscopy could help to demonstrate the presence or absence of coordination of metals with the CO or NH groups. Sure, it should be performed at higher concentrations and could be used by authors further.

Authors’ reply: Unfortunately, the NIs/metal complexes were neither isolated from solution nor crystalized, so it was not possible to obtain mass spectra or to study their structure in detail by single crystal X-ray diffraction. Concerning the recommended NMR and IR studies of the complexes – the concentrations of the samples are low (as the Reviewer correctly noticed). Additionally, the NMR spectrometer (Bruker Avance II+ 600 spectrometer) on which the ligands spectra were taken is not equipped with a special paramagnetic protocol that allows scanning of paramagnetic (Fe3+, Cu2+, Ni2+) samples.  The theoretical (DFT) modeling of the structure/binding patterns of the NIs/metal complexes was a compromise alternative. We thank the reviewer for the suggestions made and will keep them in mind for our future research.

Reviewer 3 Report

In this manuscript, Grabchev and coworkers are reporting a Napththalamide-based fluorophore system to detect metal ions in solution. Although there are several similar systems have been reported in literature, I believe this work is eventually publishable. But definitely requires a major revision with several experimental setups. Specifically following concerns must be addressed properly.

(1). Complete characterization data must be provided for all probes in the supporting information such as NMR, HRMS. Also scheme 1 should clearly indicate % yields for each step.

(2). Both absorbance and emission spectra for the probes in different solvents should be provided in supporting information. as well as for all cation tested.

(3). The number of cations tested in this approach is not sufficient. Fe(II), Al(III), Co(II) must be added.

(4). What is the counter anion for all these cation species? If authors are not keeping a consistent counter anion species, probes should be tested against anions as well.

(5). Authors must calculated LOD and binding constant by using fluorescence data.

(6). Authors should clearly describe how they calculated the fluorescence quantum yield.

(7). probe-metal complexes should be tested for stability.

(8). There are several grammatical and textual errors needs to be corrected.

Author Response

Reviewer 3:

  1. Complete characterization data must be provided for all probes in the supporting information such as NMR, HRMS. Also scheme 1 should clearly indicate % yields for each step.

Authors’ reply: The yields have been indicated in Scheme 1.

  1. Both absorbance and emission spectra for the probes in different solvents should be provided in supporting information. as well as for all cation tested.

Authors’ reply: The authors confirm that the raw data supporting the findings of this study will be available from the corresponding author, [IG], upon reasonable request.

  1. The number of cations tested in this approach is not sufficient. Fe(II), Al(III), Co(II) must be added.

Authors’ reply: NI1/NI2 sensor activity was tested on the available (and suitable) chemicals in our laboratory. The selected metal ions are with different charges (1+, 2+ and 3+), paramagnetic (Fe3+, Cu2+, Ni2+) or diamagnetic (Ag+, Zn2+, Ca2+, Mg2+) in nature.

  1. What is the counter anion for all these cation species? If authors are not keeping a consistent counter anion species, probes should be tested against anions as well.

Authors’ reply:  Аll salts tested are nitrates. The text on p.7 has been modified as follows: “In order to clarify the sensor activity of NI1 and NI2, their photophysical characteristics have been studied in the presence of metal ions with different charges: Ag+, Cu2+, Zn2+ Ca2+, Mg2+, Ni2+ and Fe3+ as nitrate salts.”

Authors must calculated LOD and binding constant by using fluorescence data.

Authors’ reply: LOD and LOQ were calculated for Fe3+ ions since the highest FE value was obtained in the presence of ferric ions for both compounds (NI1 and NI2).

Authors should clearly describe how they calculated the fluorescence quantum yield.

Authors’ reply: A short explanation has been given in section 2.5.: ”The quantum yield of fluorescence FF has been calculated on the basis of the absorption and fluorescence spectra of NI1 or NI2 by equation 1.

                                                                              (1)                                                                             

where the FF is the emission quantum yield of the sample; Fs = 0.86 is the emission quantum yield of the standard (Rhodamine 6G (Φst=0.96) and quinine bisulfate/H2SO4 1N Φst =0.546) for NI2); Ast and Au represent the absorbance of the standard and sample at the excited wavelength, respectively; while Sst and Su are the integrated emission band areas of the standard and sample, respectively, and nst and nu is the solvent refractive index of the standard and sample; subscripts u and s refer to the unknown and standard, respectively.”

  1. probe-metal complexes should be tested for stability.

Authors’ reply: The sensor/metal complexes have not been isolated in solid state. Possible structural arrangements of the ligand molecules and metal cations in the complex have been studied by DFT computations and the most likely structural organization has been proposed. The results from DFT computations suggest thermodynamic stability (in terms of free energy difference between reactants and products) of the ligand/metal complexes.

  1. There are several grammatical and textual errors needs to be corrected.

Authors’ reply: We have tried to polish the manuscript, but we would be grateful if the reviewer points out the errors noticed.

Round 2

Reviewer 3 Report

Complete characterization data must be provided for all probes in the supporting information such as NMR, HRMS. If authors claim these probes have been synthesized and tested,  It is mandatory to provide characterization HNMR 13C NMR and mass spectrometric identification data to verify the purity  of the probes. Providing just textual interpretation is not sufficient, actual NMR spectra and mass spectrometric data should be provided. I do not see this information in the supporting information. It is a mandatory to provide these characterization data while reporting any newly synthesized compounds.

Author Response

Complete characterization data must be provided for all probes in the supporting information such as NMR, HRMS. If authors claim these probes have been synthesized and tested, It is mandatory to provide characterization HNMR 13C NMR and mass spectrometric identification data to verify the purity of the probes. Providing just textual interpretation is not sufficient, actual NMR spectra and mass spectrometric data should be provided. I do not see this information in the supporting information. It is a mandatory to provide these characterization data while reporting any newly synthesized compounds.

NMR spectra of NI1 and NI2 have been added to the SI. In this study, the purity of both products was monitored by elemental analysis data and thin-layer chromatography.

Round 3

Reviewer 3 Report

Accept in present form,.

Author Response

Many thanks for your comments aimed at improving the quality of the manuscript.